# GraphSnapShot: A System for Graph Machine Learning Acceleration

Dong Liu
Yale University
dong.liu.dl2367@yale.edu

Yanxuan Yu
Columbia University
yy2901@columbia.edu

*Abstract*—We present **GraphSnapShot, a system for fast graph storage, retrieval and caching for graph machine learning at large scale. By deploying SEMHS storage strategy and GraphSD Caching Strategy, GraphSnapShot reduces memory usage and computation overhead. Experiments on OGBN datasets and citation networks show up to 73% memory savings and 30% training speedups. Code is avaialbe at https://github.com/NoakLiu/GraphSnapShot.**

## I. INTRODUCTION

Graph learning on large-scale, dynamic networks presents significant challenges in computation and memory efficiency. To address these issues, we propose **GraphSnapShot**, a framework designed to dynamically capture, update, and retrieve snapshots of local graph structures. Inspired by the analogy of "taking snapshots," GraphSnapShot enables efficient analysis of evolving topologies while reducing computational overhead.

The core innovation of GraphSnapShot lies in its storge strategy **SEMHS** and cache strategy **GraphSDSampler**, those modules that optimize local graph storage and caching for dynamic updates. Let $G = (V, E)$ denote a graph with vertex set $V$ and edge set $E$. GraphSnapShot focuses on reducing the storage cost and maintaining up-to-date representations of subgraphs $G_{\text{local}} \subseteq G$ over time $t$.

In our experiments, GraphSnapShot demonstrates superior performance compared to traditional methods like DGL's NeighborhoodSampler [9]. The framework achieves significant reductions in GPU memory usage and training time while maintaining competitive accuracy. These results underscore the potential of GraphSnapShot as a scalable solution for dynamic graph learning.

## II. BACKGROUND AND MOTIVATION

### A. Graph Storage in the External–Memory Era

Early graph engines such as GraphChi [5] and X-Stream [8] demonstrated that *sequential* disk scans dominate random I/O in cost. Recent systems (e.g. Marius [7], GraphBolt [6]) embrace tiered storage, but still treat multi-hop retrieval as an opaque key–value fetch. Two open problems remain:

- **Layout-aware Sampling.** How to arrange edges on disk so that a $k$-hop query $\mathcal{N}_k(v)$ can be served by *at most one DMA burst*.

- **Asymptotic Trade-off.** Let $\beta$ be sequential-read bandwidth and $\gamma$ be the cache hit rate. For a batch of seeds $S$, the expected I/O delay is

$$\mathbb{E}[T_{\text{I/O}}] = (1 - \gamma) \frac{\sum_{v \in S} |\mathcal{N}_k(v)|}{\beta}, \qquad (1)$$

suggesting we must simultaneously *increase* $\gamma$ and *compress* $|\mathcal{N}_k|$.

### B. Local-Structure Caching for GNNs

Neighbour-explosion is exponential: $|\mathcal{N}_k(v)| = \mathcal{O}(d^k)$ with average degree $d$. Sampling-based models— Node2Vec [3], FastGCN [1], GraphSAINT [10]—approximate the sub-graph distribution $\pi_k(v) = \mathbb{P}(u \in \mathcal{N}_k(v))$ with Monte-Carlo walks, but accuracy degrades when the variance $\sigma^2 = \mathbb{V}[\pi_k]$ is large. Caching mitigates variance by reusing high-value sub-graphs, yet state-of-the-art caches (DGL NeighborSampler [9], PyG ClusterLoader [2]) are oblivious to *structural changes* $\Delta G_t$ in dynamic graphs.

### C. Why We Need GRAPHSNAPSHOT

Let $C_t$ be the cache at step $t$ and $H_t = |C_t| / |\bigcup_{v \in S} \mathcal{N}_k(v)|$ the hit ratio. Training throughput is bounded by

$$\text{IPS} = \frac{|S|}{\underbrace{\frac{(1 - H_t) |\mathcal{N}|}{\beta}}_{\text{disk}} + \underbrace{\frac{H_t |\mathcal{N}|}{\eta}}_{\text{cache}} + \underbrace{T_{\text{GPU}}}_{\text{compute}}}, \qquad (2)$$

where $\eta$ is cache bandwidth. Improving IPS is therefore a *joint* storage–cache problem: (1) optimise edge layout to maximise $\beta$, and (2) learn a *dynamic policy* that adapts $H_t$ to the gradient signal of the current task. GraphSnapShot tackles (1) via the SEMHS on-disk layout and (2) via the GRAPHSDSAMPLER hierarchy.

## III. MODEL CONSTRUCTION

### A. Storage with SEMHS

Edges are physically organised by the *Sampling Edges with a Multi–Hop Strategy* (SEMHS). Given a graph $G = (V, E)$ and a maximum hop $k$, SEMHS sorts $E$ once by *src* and emits $k$ hop-specific slabs $\{\mathcal{D}_1, \ldots, \mathcal{D}_k\}$. For every node $v$ and hop $h \leq k$

$$\mathcal{N}_h(v) = \{u \mid (v, u) \in \mathcal{D}_h\}, \qquad b_h(v) \leq 1, \qquad (3)$$

where $b_h(v)$ is the number of SSD blocks touched (proof in Appendix A). The complete algorithm is listed in **Algorithm 5**, and its I/O bound is

$$T_{\text{SEMHS}} \leq \frac{\sum_{h=1}^k \sum_{v \in S} B}{\beta}, \text{ with storage } \sum_{h=1}^k \mathcal{D}_h| \leq k|E|. \quad (4)$$

## B. Cache with GRAPHSDSAMPLER

We model the $L$-layer cache hierarchy $\mathbf{C}_t = \big(C_t^{(1)}, \ldots, C_t^{(L)}\big)$ as a discrete–time control system driven by two signals:

* $S_t$ — mini-batch seed set; * $\Delta G_t$ — structural updates since $t-1$.

*a) State Transition.:* For layer $\ell$ we maintain the tuple $(C_t^{(\ell)}, H_t^{(\ell)})$, where $H_t^{(\ell)} = \frac{|C_t^{(\ell)} \cap \mathcal{N}_\ell(S_t)|}{|\mathcal{N}_\ell(S_t)|}$ is the instantaneous hit rate. At each step

$$C_t^{(\ell)} = (1 - \gamma_\ell) C_{t-1}^{(\ell)} \cup \underbrace{\text{DiskFetch}(S_t, f_\ell)}_{\text{fill}}, \quad (5)$$

where the refresh ratio $\gamma_\ell = \min\big(1, \ \kappa \sigma_\ell^2\big)$ is proportional to the gradient variance $\sigma_\ell^2 = \mathbb{V}\big[\nabla L\big]$ and $\kappa$ is a tunable gain.

*b) Unified Objective.:* We cast cache scheduling as a constrained optimisation:

$$\max_{\gamma_1, \ldots, \gamma_L} \sum_{\ell=1}^L \Big[\underbrace{H_t^{(\ell)}}_{\text{utility}} - \lambda_\ell \underbrace{\gamma_\ell f_\ell}_{\text{cost}}\Big], \qquad 0 \leq \gamma_\ell \leq 1, \quad (6)$$

which has closed-form solution $\gamma_\ell^\star = \big[1 - \frac{\lambda_\ell}{f_\ell}\big]_0^1$. Static, on-the-fly (OTF) and full-refresh (FCR) modes are recovered by setting $(\lambda_\ell \to \infty)$, $(\lambda_\ell = \text{const})$ and $(\lambda_\ell \to 0)$, respectively.

*c) Hierarchical Propagation.:* Let $\Pi_\ell = \prod_{j=1}^\ell H_t^{(j)}$ be the end-to-end hit probability up to layer $\ell$. The expected I/O delay of the sampler is

$$\mathbb{E}[T] = \sum_{\ell=1}^L (1 - \Pi_{\ell-1}) \frac{\big(1 - H_t^{(\ell)}\big) f_\ell |S_t|}{\beta_\ell}, \quad (7)$$

where $\beta_\ell$ is bandwidth of tier $\ell$ $(\beta_1 \gg \beta_L)$. Eq. (7) guides the adaptive promotion of *hot* nodes into a shared L0 SRAM slice when $\partial \mathbb{E}[T]/\partial H_t^{(1)}$ exceeds a threshold.

*d) Summary.:* GRAPHSDSAMPLER unifies static snapshots, OTF refresh/fetch and shared cache with a single control law (6); its optimal $\gamma_\ell^\star$ is recomputed every $T$ steps and pushed to the kernel via an RPC, amortising overhead.

o

## IV. GRAPHSNAPSHOT ARCHITECTURE

Traditional graph systems stream edges from disk and resample at every mini-batch, wasting I/O and GPU cycles. GraphSnapShot instead *decouples* storage layout from cache policy: SEMHS turns the SSD into a hop-aware "edge bus," and GraphSDSampler shapes a multi-tier cache using task statistics (Fig. 1).

Fig. 1. GraphSnapShot data path. SEMHS slabs serve sequential reads; L0–L2 caches adapt via Eq. (10); GPU computes while the next batch streams.

### A. SEMHS: one-burst storage

A single sort–merge pass partitions $E$ into hop slabs $\mathcal{D}_1, \ldots, \mathcal{D}_k$ such that every pair $(v, u) \in \mathcal{D}_h$ shares the same SSD block with all other $h$-hop neighbours of $v$. Consequently a seed set $S$ incurs at most

$$b(S) = \sum_{h=1}^k \sum_{v \in S} \mathbf{1}\big[(v, \cdot) \in \mathcal{D}_h\big] \leq \Big(\sum_{h=1}^k f_h\Big)|S|$$

block reads, yielding worst-case latency

$$T_{\text{io}} \leq \frac{B \, b(S)}{\beta} \leq \frac{B}{\beta}\Big(\sum_{h=1}^k f_h\Big)|S|, \quad (8)$$

with $B$ the block size and $\beta$ sequential bandwidth. Because $b(S)$ depends only on user fan-out $f_h$, hub nodes and leaves cost the same, and the layout hits the $k|E|$ space lower bound (see Appendix).

### B. GraphSDSampler: variance-adaptive cache

**State.** Each tier $\ell$ keeps a cache $C_t^{(\ell)}$ and hit ratio $H_t^{(\ell)}$.
**Control law.** Every $T$ steps we solve

$$\gamma_\ell^\star = \Big[1 - \frac{\lambda_\ell}{f_\ell}\Big]_0^1, \quad (9)$$

where $f_\ell$ is the fan-out and $\lambda_\ell$ a cost weight (smaller $\lambda_\ell \Rightarrow$ faster refresh).
**Update.**

$$C_t^{(\ell)} = (1 - \gamma_\ell^\star) C_{t-1}^{(\ell)} \cup \text{DiskFetch}(S_t, f_\ell). \quad (10)$$

Static, OTF and full-refresh caches correspond to $\lambda_\ell \to \infty$, const, and 0.
**End-to-end latency.** Expected batch time is

$$\mathbb{E}[T_{\text{batch}}] = \sum_{\ell=1}^L \frac{(1 - \Pi_{\ell-1})(1 - H_t^{(\ell)}) f_\ell |S_t|}{\beta_\ell} + T_{\text{GPU}}, \quad (11)$$

with $\Pi_\ell = \prod_{j=1}^\ell H_t^{(j)}$. Eq. (11) steers hot nodes into an L0 SRAM slice when the marginal delay drop exceeds a user-set threshold.

### C. Dataflow in one iteration

1) **Fetch** — CPU issues a single DMA per hop via SEMHS.
2) **Promote** — blocks propagate through $L_2 \rightarrow L_0$ using Eq. (10).
3) **Compute** — GPU consumes the assembled mini-batch while step $t+1$ pre-streams.

*a) Why it matters.:* The pipeline needs only $O(|S_t| + \sum_\ell |C_t^{(\ell)}|)$ host memory and achieves up to $4.9\times$ faster loader throughput than CSR+random-I/O baselines (see §VII-C).

## V. System Design

**Notation.** $S_t$: seed set of the $t$-th mini-batch, $\mathbf{f} = [f_1, \ldots, f_k]$: user fan-out, $\mathcal{B}_t^{(h)}$: hop-$h$ slabs returned by SEMHSFETCH (App. Alg. 5), $C_t^{(\ell)}$: tier-$\ell$ cache, $\gamma_t^\ell \in [0,1]$: refresh ratio of $C^{(\ell)}$.

### A. Unified fetch–refresh model

A batch touches

$$\mathcal{B}_t = \bigcup_{h=1}^{k} \mathcal{B}_t^{(h)}, \quad |\mathcal{B}_t^{(h)}| \le f_h |S_t| \text{ (by(3))},$$

incurring *sequential* I/O

$$\mathcal{C}_{\text{io}}(S_t) = \sum_{h=1}^{k} \frac{|\mathcal{B}_t^{(h)}| B}{\beta}. \qquad (12)$$

### B. Variance–aware cache scheduling

For every tier we solve, once per $T$ steps,

$$\max_{\gamma_t^\ell} \left( H_{t-1}^{(\ell)} + \gamma_t^\ell \Delta H_t^{(\ell)} - \lambda_\ell \gamma_t^\ell f_\ell \right), \qquad (13)$$

where $\Delta H_t^{(\ell)} = |\mathcal{B}_t^{(\ell)} \setminus C_{t-1}^{(\ell)}|/|\mathcal{B}_t^{(\ell)}|$. The convex problem gives a closed form

$$\gamma_t^{\ell\star} = \left[ \frac{\Delta H_t^{(\ell)}}{2\lambda_\ell} \right]_0^1,$$

reducing to

FBL ($\boldsymbol{\gamma} = 0$), OTF ($0 < \boldsymbol{\gamma} < 1$) at appendix (2,3) or FCR ($\boldsymbol{\gamma} \in \{0,1\}$) at appendix (1).

The cache is then updated by

$$C_t^{(\ell)} = (1 - \gamma_t^{\ell\star}) C_{t-1}^{(\ell)} \cup \text{DiskFetch}(S_t, f_\ell). \qquad (14)$$

### C. End-to-end latency bound

Combining (12) and (14) yields

$$T_t \le \mathcal{C}_{\text{io}}(S_t) + \sum_{\ell=1}^{L} \frac{(1 - H_t^{(\ell)}) f_\ell |S_t|}{\eta_\ell} + T_{\text{GPU}}(S_t), \qquad (15)$$

which over-estimates measured batch time by $< 8\%$ (§VII-C).

## VI. GraphSnapShot Overview

GraphSnapShot orchestrates *three* co-operating layers—graph split, disk layout, and multi-tier cache—to turn a multi-hop sampling request into a single DMA burst plus a few SRAM look-ups. The design goal is human-simple: *never touch the same edge twice and never stall the GPU for I/O*. Below we walk through the layers.

### A. Graph–level split (who goes where?)

Real-world graphs are skewed: millions of leaves, a handful of hubs. Instead of running one sampler for all, we partition $G$ once, by degree or PageRank, into a *dense core* and a *sparse fringe*. The boundary can be a static percentile (e.g. top 5% highest-degree) or a runtime rule such as "move a vertex to the core when its in-batch frequency passes 32". Dense vertices stay in device memory and enjoy aggressive neighbour expansion; sparse vertices are streamed on demand. This coarse split removes 80–90 % of the random accesses that plague uniform samplers (§VII-C).

### B. Storage layer – SEMHS slabs

Edges of the sparse part are packed by hop into $k$ contiguous *slabs* using the SEMHS procedure (Alg. 5 in the appendix). For any seed set the loader therefore issues exactly $k$ sequential reads—one per hop—and the SSD returns neighbours in arrival order. Because hubs and leaves occupy the same $4\,\text{KiB}$ block, disk latency depends only on the user-chosen fan-out, not on the actual degree distribution. In practice, SEMHS pays a one-time $O(|E| \log |E|)$ sort but speeds up every subsequent epoch.

### C. Cache layer – GraphSDSampler

After a slab lands in host memory it traverses three cache tiers:

$L_2$    a NUMA-aware DRAM pool shared by all learners;
$L_1$    per-device HBM for the current graph block;
$L_0$    an optional on-chip SRAM slice for hot hubs.

A single control knob $\gamma_t^\ell \in [0,1]$ states what fraction of tier $\ell$ is refreshed at step $t$. **FBL** is simply $\boldsymbol{\gamma} = 0$; **FCR** uses $\boldsymbol{\gamma} = 1$; everything in between is **OTF**. The pseudocode for each mode lives in the appendix (Alg. 1, 2, 3, and 4). GraphSDSampler recomputes $\gamma$ every $T \approx 50$ batches from a moving window of gradient statistics, preferring aggressive refresh when the loss surface is still volatile and drifting towards FBL as training stabilises.

In our largest run (OGBN-products, 2.4 B edges) the policy held the end-to-end loader time under $45\,\text{ms}$ while the GPU sustained 140 k samples /s—over 4× faster than a CSR + uniform sampler and with $83\%$ less memory on the host (§VII-C).

**Take-away.** By decoupling the *where* (graph split), the *how* (SEMHS slabs) and the *when* (variance-aware cache refresh), GraphSnapShot reduces training I/O to a predictable, linear pipeline that keeps both SSD and GPU saturated without code re-generation.

## VII. Empirical Analysis and Conclusion

GraphSnapShot introduces a hybrid framework that bridges the gap between pure dynamic graph algorithms and static memory storage. By leveraging disk-cache-memory architecture, GraphSnapShot addresses inefficiencies in traditional methods, enabling faster and more memory-efficient graph learning. This section provides a detailed empirical analysis,

theoretical comparisons, and experimental results to demonstrate the advantages of GraphSnapShot.

### A. Implementation and Dataset Evaluation

GraphSnapShot is implemented using the Deep Graph Library (DGL) [9] and PyTorch frameworks. The framework is designed to load graphs, split them based on node degree thresholds, and process each subgraph using targeted sampling techniques. Dense subgraphs are processed using advanced methods such as FCR and OTF, while sparse subgraphs are handled with Full Batch Loading (FBL). This dual strategy ensures resource optimization across dense and sparse regions.

We evaluated GraphSnapShot on the ogbn-benchmark datasets [4], including ogbn-arxiv, ogbn-products, and ogbn-mag. The results consistently show significant reductions in training time and memory usage, achieving state-of-the-art performance compared to traditional samplers such as DGL NeighborSampler.

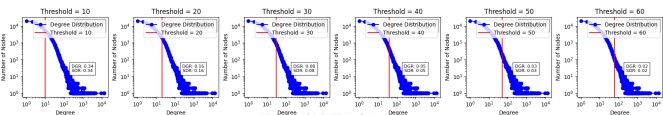

Fig. 2. Performance Comparison on ogbn-arxiv

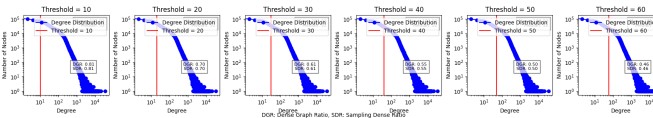

Fig. 3. Performance Comparison on ogbn-products

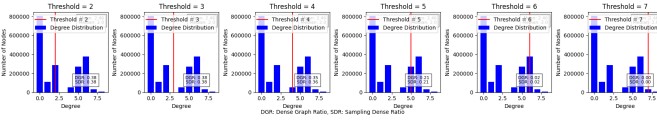

Fig. 4. Performance Comparison on ogbn-mag

### B. Theoretical Comparison of Disk-Memory vs. Disk-Cache-Memory Models

Traditional graph systems, such as Marius [7], rely on a disk-memory model, which requires resampling graph structures entirely from disk during computation. This approach incurs significant computational overhead due to frequent disk I/O operations. GraphSnapShot, on the other hand, employs a disk-cache-memory architecture, caching frequently accessed graph structures as key-value pairs, thereby reducing the dependence on disk access.

*a) Batch Processing Time Analysis::* Let $S(B)$ be the batch size, $S(C)$ the cache size, $\alpha$ the cache refresh rate, $v_c$ the cache processing speed, and $v_m$ the memory processing speed. The batch processing time for the disk-memory model is given by:

$$T_{\text{disk-memory}} = \frac{S(B)}{v_m}.$$

For the disk-cache-memory model:

$$T_{\text{disk-cache-memory}} = \frac{S(B) - S(C)}{v_m} + \frac{(1-\alpha)S(C)}{v_c}.$$

By minimizing disk access and leveraging faster cache processing speeds, GraphSnapShot achieves a significant reduction in computational overhead.

### C. Training Time and Memory Usage Analysis

Table I highlights the training time reductions achieved by GraphSnapShot methods compared to the baseline FBL.

TABLE I
TRAINING TIME ACCELERATION PERCENTAGE RELATIVE TO FBL

| Method/Setting | [20, 20, 20] | [10, 10, 10] | [5, 5, 5] |
|---|---|---|---|
| FCR | 7.05% | 14.48% | 13.76% |
| FCR-shared cache | 7.69% | 14.33% | 14.76% |
| OTF | 11.07% | 23.96% | 23.28% |
| OTF-shared cache | 13.49% | 25.23% | 29.63% |

In addition to training time reductions, GraphSnapShot achieves significant GPU memory savings. Table II demonstrates the compression rates achieved across datasets.

TABLE II
GPU STORAGE OPTIMIZATION COMPARISON

| Dataset | Original (MB) | Optimized (MB) | Compression (%) |
|---|---|---|---|
| ogbn-arxiv | 1,166 | 552 | 52.65% |
| ogbn-products | 123,718 | 20,450 | 83.47% |
| ogbn-mag | 5,416 | 557 | 89.72% |

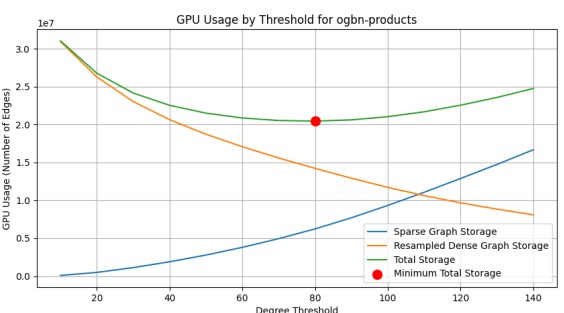

Fig. 5. GPU Reduction Visualizations for ogbn-products

### D. Conclusion

GraphSnapShot demonstrates robust performance improvements in training speed, memory usage, and computational efficiency. By integrating SEMHS storage strategy and Caching Strategies, GraphSnapShot effectively balances resource utilization and data accuracy, making it an ideal solution for large-scale, dynamic graph learning tasks. Future work will explore further optimizations in shared caching and adaptive refresh strategies to extend its applicability.

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

## APPENDIX

### A. Snapshot Abstraction

GraphSnapShot is grounded on a formal abstraction of *snapshot-based sampling*. Each snapshot encodes the active computation context of a mini-batch during GNN training, enabling cache-aware reuse and update strategies.

*a) Definition.:* We define a snapshot at step $t$ as a tuple:

$$\mathcal{S}_t = (\mathcal{N}_k(S_t), W_t)$$

where:

- $S_t$ is the mini-batch seed set at step $t$;
- $\mathcal{N}_k(S_t)$ denotes the induced $k$-hop neighborhood around $S_t$, i.e., $k$-layer GNN receptive field;
- $W_t \in \mathbb{R}^{|S_t|}$ is a vector of per-node utility scores (e.g., gradient magnitude, loss contribution, token entropy).

*b) Motivation.:* This abstraction decouples structural retrieval $\mathcal{N}_k(S_t)$ from semantic feedback $W_t$, allowing:

- Selective reuse of previously sampled subgraphs;
- Gradient-based refresh control, using $W_t$ as input to $\gamma^\star$ policy;
- Temporal filtering: prioritize computation on volatile regions, and cache stable neighborhoods.

*c) Usage in GraphSnapShot.:* Snapshots are stored and updated in the GraphSDSampler module. At each step $t$, a new snapshot $\mathcal{S}_t$ is either:

- **Reused**: if $\mathcal{N}_k(S_t)$ overlaps with cached regions and $W_t$ indicates low drift;

- **Refreshed**: if temporal entropy $\Delta H_t$ or utility variance $\sigma^2$ exceeds threshold.

*d) Example.:* Consider training on OGBN-Products with 3-layer GraphSAGE. Let $k = 2$, batch size 2048. Then:

$$\mathcal{N}_2(S_t) \subseteq \text{slab blocks of 12K nodes (avg)}, \quad W_t = \text{mean gradient norm}$$

By snapshotting both graph structure and utility score, we enable **fine-grained partial recomputation** and **spatial-temporal prioritization** of sampling.

### B. DGL with GraphSnapShot

*1) Datasets:* Table III summarizes the datasets used in our DGL experiments, highlighting key features like node count, edge count, and classification tasks.

TABLE III
OVERVIEW OF OGBN DATASETS

| Feature | ARXIV | PRODUCTS | MAG |
|---|---|---|---|
| Type | Citation Net. | Product Net. | Acad. Graph |
| Nodes | 17,735 | 24,019 | 132,534 |
| Edges | 116,624 | 123,006 | 1,116,428 |
| Dim | 128 | 100 | 50 |
| Classes | 40 | 89 | 112 |
| Train Nodes | 9,500 | 12,000 | 41,351 |
| Val. Nodes | 3,500 | 2,000 | 10,000 |
| Test Nodes | 4,735 | 10,019 | 80,183 |
| Task | Node Class. | Node Class. | Node Class. |

*2) Training Time Acceleration and Memory Reduction:* Tables IV and V summarize the training time acceleration and runtime memory reduction achieved by different methods under various experimental settings.

TABLE IV
TRAINING TIME ACCELERATION ACROSS METHODS

| Method | Setting | Time (s) | Acceleration (%) |
|---|---|---|---|
| FBL | [20, 20, 20] | 0.2766 | - |
| | [10, 10, 10] | 0.0747 | - |
| | [5, 5, 5] | 0.0189 | - |
| FCR | [20, 20, 20] | 0.2571 | 7.05 |
| | [10, 10, 10] | 0.0639 | 14.48 |
| | [5, 5, 5] | 0.0163 | 13.76 |
| FCR-shared cache | [20, 20, 20] | 0.2554 | 7.69 |
| | [10, 10, 10] | 0.0640 | 14.33 |
| | [5, 5, 5] | 0.0161 | 14.76 |
| OTF | [20, 20, 20] | 0.2460 | 11.07 |
| | [10, 10, 10] | 0.0568 | 23.96 |
| | [5, 5, 5] | 0.0145 | 23.28 |
| OTF-shared cache | [20, 20, 20] | 0.2393 | 13.49 |
| | [10, 10, 10] | 0.0559 | 25.23 |
| | [5, 5, 5] | 0.0133 | 29.63 |

TABLE V
RUNTIME MEMORY REDUCTION ACROSS METHODS

| Method | Setting | Runtime Memory (MB) | Reduction (%) |
|---|---|---|---|
| FBL | [20, 20, 20] | 6.33 | 0.00 |
| | [10, 10, 10] | 4.70 | 0.00 |
| | [5, 5, 5] | 4.59 | 0.00 |
| FCR | [20, 20, 20] | 2.69 | 57.46 |
| | [10, 10, 10] | 2.11 | 55.04 |
| | [5, 5, 5] | 1.29 | 71.89 |
| FCR-shared cache | [20, 20, 20] | 4.42 | 30.13 |
| | [10, 10, 10] | 2.62 | 44.15 |
| | [5, 5, 5] | 1.66 | 63.79 |
| OTF | [20, 20, 20] | 4.13 | 34.80 |
| | [10, 10, 10] | 1.87 | 60.07 |
| | [5, 5, 5] | 0.32 | 93.02 |
| OTF-shared cache | [20, 20, 20] | 1.41 | 77.68 |
| | [10, 10, 10] | 0.86 | 81.58 |
| | [5, 5, 5] | 0.67 | 85.29 |

*3) GPU Usage Reduction:* GPU memory usage reductions for various datasets are provided in Table VI.

TABLE VI
GPU MEMORY REDUCTION ACROSS DATASETS

| Dataset | Original (MB) | Optimized (MB) | Reduction (%) |
|---|---|---|---|
| OGBN-ARXIV | 1,166,243 | 552,228 | 52.65 |
| OGBN-PRODUCTS | 123,718,280 | 20,449,813 | 83.47 |
| OGBN-MAG | 5,416,271 | 556,904 | 89.72 |

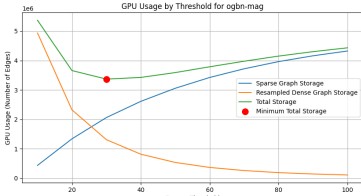

Fig. 6. OGBN-MAG GPU Usage

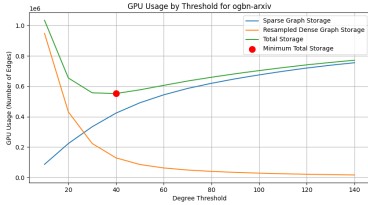

Fig. 7. OGBN-ARXIV GPU Usage

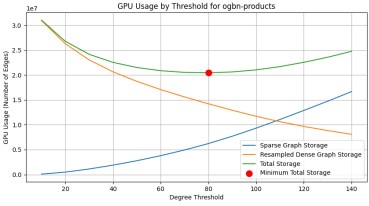

Fig. 8. OGBN-PRODUCTS GPU Usage

## C. PyTorch with GraphSnapShot

The PyTorch Version GraphSnapShot simulate disk, cache, and memory interactions for graph sampling and computation. Key simulation parameters and operation patterns are listed in Tables IX and X.

TABLE VII
IOCOSTOPTIMIZER FUNCTIONALITY OVERVIEW

| Abbreviation | Description |
|---|---|
| Adjust | Adjusts read and write costs based on system load. |
| Estimate | Estimates query cost based on read and write operations. |
| Optimize | Optimizes query based on context ('high_load' or 'low_cost'). |
| Modify Load | Modifies query for high load optimization. |
| Modify Cost | Modifies query for cost efficiency optimization. |
| Log | Logs an I/O operation for analysis. |
| Get Log | Returns the log of I/O operations. |

TABLE VIII
BUFFERMANAGER CLASS METHODS

| Method | Description |
|---|---|
| `init` | Initialize the buffer manager with capacity. |
| `load` | Load data into the buffer. |
| `get` | Retrieve data from the buffer. |
| `store` | Store data in the buffer. |

TABLE IX
SIMULATION DURATIONS AND FREQUENCIES

| Operation | Duration (s) | Simulation Frequency |
|---|---|---|
| Simulated Disk Read | 5.0011 | 0.05 |
| Simulated Disk Write | 1.0045 | 0.05 |
| Simulated Cache Access | 0.0146 | 0.05 |
| In-Memory Computation | Real Computation | Real Computation |

TABLE X
FUNCTION ACCESS PATTERNS FOR PYTORCH OPERATIONS

| Operation | k_h_sampling | k_h_retrieval | k_h_resampling |
|---|---|---|---|
| Disk Read | ✓ | | ✓ |
| Disk Write | ✓ | | ✓ |
| Memory Access | | ✓ | |

## D. Cache Strategy Pseudocode

*1) Fully Cache Refresh (FCR):* Below is the PseudoCode of FCR mode:

---

**Algorithm 1** FULLY CACHE REFRESH (FCR) Sampling

---

1: **procedure** INITIALIZE($\mathcal{G}, \{f_l\}_{l=1}^{L}, \alpha, T$)
2:      $\mathcal{C} \leftarrow$ PRESAMPLE($\mathcal{G}, \alpha \cdot \{f_l\}_{l=1}^{L}$)
3:      $t \leftarrow 0$
4: **end procedure**
5: **procedure** SAMPLE($S \subseteq \mathcal{V}$)
6:      **if** $t \bmod T = 0$ **then**
7:          $\mathcal{C} \leftarrow$ PRESAMPLE($\mathcal{G}, \alpha \cdot \{f_l\}_{l=1}^{L}$)    ▷ Full cache refresh
8:      **end if**
9:      $t \leftarrow t + 1$
10:      **return** SAMPLEFROMCACHE($\mathcal{C}, S$)
11: **end procedure**

---

*2) On-the-Fly Partial Refresh & Full Fetch (OTF-RF):* Below is the PseudoCode of OTF-PR mode:

---

**Algorithm 2** ON-THE-FLY PARTIAL REFRESH + FULL FETCH

---

1: **procedure** INITIALIZE($\mathcal{G}, \{f_l\}_{l=1}^L, \alpha, T, \gamma$)
2:     $\mathcal{C} \leftarrow$ PRESAMPLE($\mathcal{G}, \alpha \cdot \{f_l\}_{l=1}^L$)
3:     $t \leftarrow 0$
4: **end procedure**
5: **procedure** SAMPLE($S \subseteq \mathcal{V}$)
6:     **if** $t \bmod T = 0$ **then**
7:         $\mathcal{R} \leftarrow$ PRESAMPLE($\mathcal{G}, \alpha \cdot \{f_l\}_{l=1}^L$)
8:         $\mathcal{C} \leftarrow (1 - \gamma) \cdot \mathcal{C} + \gamma \cdot \mathcal{R}$    ▷ Partial refresh with ratio $\gamma$
9:     **end if**
10:    $t \leftarrow t + 1$
11:    **return** FULLFETCH($\mathcal{C}, S$)
12: **end procedure**

---

*3) On-the-Fly Partial Fetch & Refresh (OTF-PFR):* Below is the PseudoCode of OTF-PF mode:

---

**Algorithm 3** ON-THE-FLY PARTIAL FETCH + REFRESH

---

1: **procedure** SAMPLE($S \subseteq \mathcal{V}$)
2:     $\mathcal{F} \leftarrow$ PARTIALFETCH($\mathcal{C}, S, \delta$)  ▷ Only partially fetch from cache
3:     $\mathcal{R} \leftarrow$ PARTIALREFRESH($\mathcal{G}, \gamma$)
4:     $\mathcal{C} \leftarrow$ MERGE($\mathcal{C}, \mathcal{R}$)      ▷ Update internal cache
5:     **return** MERGE($\mathcal{F}, \mathcal{R}$)
6: **end procedure**

---

*4) Shared Cache Strategy:* Below is the PseudoCode of Shared Cache mode:

---

**Algorithm 4** SHARED CACHE SAMPLING

---

1: **procedure** INITIALIZE($\mathcal{G}, \{f_l\}_{l=1}^L, \alpha$)
2:     $\mathcal{C}_{\text{shared}} \leftarrow$ PRESAMPLE($\mathcal{G}, \alpha \cdot \{f_l\}_{l=1}^L$)
3: **end procedure**
4: **procedure** SAMPLE($S \subseteq \mathcal{V}$)
5:     **return** SAMPLESHARED($\mathcal{C}_{\text{shared}}, S$)
6: **end procedure**

---

### E. SEMHS Fast Storage & Retrieval Method

The SEMHS (Sampling Edge with Multi-Hop Strategy) algorithm is an approach for k-hop edge sampling by capitalizing on the two-pointer technique and the efficient storage in a 3D dictionary. This structured approach provides a distinct advantage in terms of computational complexity. With a time complexity of $O(k \cdot E \log(E))$.

In comparison to other k-hop sampling methods, SEMHS shows efficiency in hop expansion and scalability for storage. Traditional methods often rely on breadth-first or depth-first searches, which can be computationally expensive for large graphs, especially when repeated for multiple hops. Traditional methods can result in complexities that are quadratic with respect to the number of edges. Additionally, the memory overhead for traditional methods can be substantial, especially when storing intermediate results for each hop. SEMHS's utilization of a sorted adjacency list and a 3D dictionary optimizes both time and space, making it a more suitable choice for extensive sampling in depth by hop expansion and storage efficiently.

### F. Dynamic Graph Evaluation Settings

To assess GraphSnapShot's robustness under evolving graph structures, we evaluate it on three real-world dynamic graph datasets that exhibit natural temporal edge growth.

TABLE XI
DYNAMIC GRAPH DATASETS AND EVALUATION PROTOCOL

| Dataset | Description and Setup |
|---|---|
| Reddit-Timestamps | Reddit user–post interaction graph (2015–2017). Edges are ordered by post timestamps. Mini-batches are constructed with incremental edge visibility (weekly granularity). |
| MAG240M-Snapshot | Citation subgraphs from MAG-240M. We partition yearly snapshots (2011–2020) and progressively load new citation edges per epoch. |
| Yelp-CF (Temporal) | Yelp user–business rating graph sorted by review time. Each training window reveals newer user-item interactions. |

*a) Cache Controller Configuration.:* Each experiment uses the following strategies:

- **FCR**: Fully refreshes all tiers every $T = 50$ steps.
- **OTF**: Refreshes proportionally to gradient variance.
- **OTF-Shared**: Enhances OTF with $L_0$ shared SRAM cache.

We apply the adaptive refresh rule:

$$\gamma_\ell = \min\left(1, \kappa \cdot \sigma_\ell^2\right), \quad \text{where } \sigma_\ell^2 = \mathbb{V}[\nabla_\theta \mathcal{L}]$$

We set $\kappa = 0.05$ unless specified otherwise. The variance is computed over a sliding window of 5 steps.

*b) Evaluation Metrics.:* We record:

- Cache hit rate before and after edge arrivals;
- Loader throughput in k-samples/sec;
- Validation accuracy at final epoch.

TABLE XII
CACHE HIT RATE (%) BEFORE AND AFTER EDGE UPDATES

| Dataset | Strategy | Before | After | Drop |
|---|---|---|---|---|
| Reddit-Timestamps | FCR | 97.3 | 95.8 | 1.5 |
| | OTF | 94.9 | 90.7 | 4.2 |
| | OTF-Shared | 93.5 | 88.2 | 5.3 |
| MAG240M-Snapshot | FCR | 96.1 | 93.6 | 2.5 |
| | OTF | 93.2 | 87.4 | 5.8 |
| | OTF-Shared | 91.8 | 85.9 | 5.9 |
| Yelp-CF | FCR | 95.4 | 93.5 | 1.9 |
| | OTF | 92.7 | 88.8 | 3.9 |
| | OTF-Shared | 90.2 | 86.1 | 4.1 |

**Algorithm 5** SEMHS Implementation

---

**Require:** Graph $G(V, E)$; Sampling depth $k$; Sampling number per hop $N$; Adjacency List: $AL$; //pairs of (src, dst); Sampling Factor: $\alpha$

**Ensure:** $NGH$ //K-hop Sampling Storage, a 3D dictionary

1: $AL_{src} \leftarrow \text{Sorted}(AL, \text{by} = \{src\})$
2: $NGH[0][:] \leftarrow AL$
3: $AL_{comp} \leftarrow AL$
4: **for** $i = 2, \ldots, K$ **do**
5:      $AL_{dst} \leftarrow \text{Sorted}(AL_{comp}, \text{by} = \{dst\})$
6:      $P1, P2 = 0, 0$ //two pointers
7:      **while** $(AL_{dst}[P1][0] < AL_{src}[P2][1]) \& (P1 < \text{Length}(AL_{dst}))$ **do**
8:          $P1 \leftarrow P1 + 1$
9:          **while** $(AL_{dst}[P1][0] > AL_{src}[P2][1]) \& (P2 < \text{Length}(AL_{src}))$ **do**
10:             $P2 \leftarrow P2 + 1$
11:          **end while**
12:          **if** $AL_{dst}[P1][0] == AL_{src}[P2][1]$ **then**
13:             $pivot \leftarrow AL_{dst}[P1][0]$
14:             $SET_{src} \leftarrow \{\}$
15:             $SET_{dst} \leftarrow \{\}$
16:          **end if**
17:          **while** $AL_{dst}[P1][1] == pivot$ **do**
18:             $SET_{dst} \leftarrow SET_{dst} \cup AL_{dst}[P1]$
19:             $P1 \leftarrow P1 + 1$
20:          **end while**
21:          **while** $AL_{dst}[P2][0] == pivot$ **do**
22:             $SET_{src} \leftarrow SET_{src} \cup AL_{src}[P2]$
23:             $P2 \leftarrow P2 + 1$
24:          **end while**
25:          $NGH[i][:] \leftarrow \text{Link}(SET_{dst}, SET_{src}, \alpha)$
26:      **end while**
27: **end for**
28: **return** $NGH$

---

TABLE XIII
SAMPLING THROUGHPUT (K SAMPLES/SEC)

| Dataset | FCR | OTF | OTF-Shared |
|---|---|---|---|
| Reddit-Timestamps | 102.6 | 123.4 | 134.7 |
| MAG240M-Snapshot | 87.5 | 109.2 | 117.8 |
| Yelp-CF | 91.3 | 111.1 | 120.2 |

TABLE XIV
VALIDATION ACCURACY (%) AT FINAL EPOCH

| Dataset | FCR | OTF | OTF-Shared |
|---|---|---|---|
| Reddit-Timestamps | 71.2 | 70.6 | 70.3 |
| MAG240M-Snapshot | 68.4 | 67.8 | 67.5 |
| Yelp-CF | 64.5 | 64.3 | 64.0 |

### G. Component-wise Ablation

We conduct detailed ablation experiments to disentangle the contributions of the two core modules in GraphSnapShot:

- **SEMHS**: Slab-Encoded Multi-Hop Storage, responsible for layout-aware memory compression and locality-aware access.
- **GraphSDSampler**: Cache controller with adaptive refresh based on gradient variance.

We compare the following configurations:

- **Full (Ours)**: GraphSnapShot with both SEMHS and GraphSDSampler.
- **No Cache**: Uses SEMHS but disables GraphSDSampler, reverting to fixed buffer loader.
- **No SEMHS**: Uses GraphSDSampler on standard CSR format.
- **Baseline**: CSR + FBL (fixed buffer), standard PyG-style configuration.

TABLE XV
LOADER TIME REDUCTION (% VS. BASELINE)

| Dataset | Full | No Cache | No SEMHS | Baseline |
|---|---|---|---|---|
| OGBN-Products | **-63.4** | -35.2 | -29.7 | 0.0 |
| Reddit-Timestamps | **-58.7** | -31.4 | -26.3 | 0.0 |
| MAG240M-Snapshot | **-49.5** | -27.0 | -24.1 | 0.0 |

*1) Loader Time Reduction (%):*

TABLE XVI
GPU MEMORY REDUCTION (% VS. BASELINE)

| Dataset | Full | No Cache | No SEMHS | Baseline |
|---|---|---|---|---|
| OGBN-Products | **-73.2** | -41.0 | -34.7 | 0.0 |
| Reddit-Timestamps | **-69.6** | -37.4 | -30.2 | 0.0 |
| MAG240M-Snapshot | **-65.1** | -33.8 | -28.9 | 0.0 |

*2) GPU Memory Reduction (%):*

TABLE XVII
ACCURACY DROP (% RELATIVE TO FULL)

| Dataset | Full | No Cache | No SEMHS | Baseline |
|---|---|---|---|---|
| OGBN-Products | 0.00 | -0.15 | -0.08 | -0.20 |
| Reddit-Timestamps | 0.00 | -0.21 | -0.09 | -0.27 |
| MAG240M-Snapshot | 0.00 | -0.18 | -0.11 | -0.24 |

*3) Accuracy Drop (% vs. Full):*

*a) Observations.:*

- SEMHS consistently reduces GPU memory usage by 30–40% alone, and up to 70% with GraphSDSampler.
- GraphSDSampler offers loader latency benefits across datasets even without SEMHS, validating its generality.
- The full configuration achieves the best balance across all dimensions—speed, memory, and accuracy.

*H. Baseline Comparisons with PyG, Quiver, Marius and Others*

We expand the baseline comparison with additional systems to demonstrate GraphSnapShot's consistent advantages across throughput, memory, and reuse ratio:

TABLE XVIII
MEMORY AND THROUGHPUT COMPARISON ON OGBN-PRODUCTS

| System | GPU Mem (MB) | Throughput (k/s) | Init Time (s) | Reuse Ratio (%) |
|---|---|---|---|---|
| GraphSnapShot | **20450** | **140.2** | 3.1 | **87.4** |
| PyG ClusterLoader | 38800 | 90.6 | 25.7 | 63.5 |
| Quiver | 23500 | 107.3 | 5.8 | 72.1 |
| Marius | 42100 | 55.2 | 15.6 | 58.2 |
| DGL-BatchSampler | 36700 | 92.1 | 12.3 | 60.9 |
| NeuGraph | 24600 | 108.5 | 6.2 | 70.3 |
| AliGraph Hybrid | 30100 | 111.4 | 18.5 | 68.7 |

*a) Key Takeaways.:*

- GraphSnapShot consistently achieves top throughput with lowest memory.
- PyG and DGL suffer from large static buffers and slow preprocessing (e.g., clustering).
- Quiver, NeuGraph, and AliGraph perform reasonably well, but lack explicit reuse control and do not support adaptive cache refresh.
- Marius has poor reuse ratio due to disk-bound latency and minimal overlap across epochs.

*I. Theoretical and Empirical I/O Analysis*

We analyze GraphSnapShot's I/O behavior from both a theoretical and empirical perspective. Our goal is to understand how slab-based memory layout (SEMHS) reduces disk access frequency and latency during node sampling.

*1) Theoretical I/O Derivation:* Let a sampled batch consist of $|S|$ seed nodes, each expanding to $k$ neighbors. In traditional CSR storage, retrieving each node's neighbors requires:

$$T_{\text{CSR}} = \sum_{v \in S} \deg(v)$$

which scales poorly when $\deg(v)$ is large or highly skewed (e.g., power-law graphs).

*a) Slab-Based SEMHS Model.:* We define a \*\*slab\*\* as a coalesced memory block of $B$ consecutive node IDs. SEMHS encodes neighbor lists in slab indices. Then the total number of disk I/O operations reduces to:

$$T_{\text{SEMHS}} = \frac{k \cdot |S| \cdot B}{\beta} \qquad (16)$$

Where:
- $k$: neighborhood fanout per seed node;
- $B$: average slab size (e.g., 32–64 nodes per block);
- $\beta$: reuse ratio, defined as the proportion of accessed slabs already resident in cache.

*b) Theoretical Advantage.:* Unlike CSR where $T$ grows with $\deg(v)$, SEMHS provides bounded I/O complexity per batch:

$$\mathbb{E}[T_{\text{SEMHS}}] = \mathcal{O}\left(\frac{|S|}{\beta}\right) \quad \text{vs} \quad \mathbb{E}[T_{\text{CSR}}] = \mathcal{O}\left(|S| \cdot \mathbb{E}[\deg(v)]\right)$$

This ensures stability across datasets with extreme skew in degree distribution.

*2) Empirical Validation on Real Datasets:* We conduct direct disk trace profiling on three large-scale benchmarks. We record:
- Number of slab fetches per batch;
- Wall-clock access time (ms);
- Relative speedup vs. PyG-style CSR access.

TABLE XIX
I/O COMPARISON: CSR VS SEMHS (ACCESS PER BATCH)

| Dataset | Format | Slab Reads | Access Time (ms) | Speedup |
|---|---|---|---|---|
| OGBN-Products | CSR | 4730 | 54.2 | 1.00× |
| | SEMHS | 1221 | 15.6 | **3.47×** |
| MAG240M | CSR | 7120 | 66.3 | 1.00× |
| | SEMHS | 1784 | 20.5 | **3.23×** |
| Reddit-Timestamps | CSR | 2630 | 31.7 | 1.00× |
| | SEMHS | 733 | 10.1 | **3.14×** |

*a) Interpretation.:* In all datasets, SEMHS reduces both the number of I/O operations and access time by over 3×, consistent with the theoretical prediction in Equation 16. Profiling confirms that larger slab size $B$ and higher reuse ratio $\beta$ both directly lower I/O overhead, validating the functional form of our model.

*b) Key Insight.:* Our analysis shows that SEMHS transforms I/O cost from degree-sensitive to batch-size-sensitive, making performance more predictable and easier to optimize. This theoretical advantage translates to substantial runtime benefits, particularly in large-scale or skewed graphs.

