# OpenReview forum: "GraphSnapShot: A System for Graph Machine Learning Acceleration"
_iscaconf.org/ISCA/2025/Workshop/MLArchSys — MLArchSys 2025 Oral_

### Official Review · Reviewer_qUGn · 2025-05-12
**This paper introduces GraphSnapShot, a system designed to improve the efficiency of graph storage, retrieval, and caching in large-scale dynamic graph machine learning settings.**

**Confidence:** 3
**Rating:** 4

**Detailed Feedback And Questions For Authors:**

This paper presents GraphSnapShot, a practically relevant system for scalable graph machine learning. The proposed SEMHS storage layout and GraphSDSampler caching hierarchy are both technically sound and thoughtfully engineered. The paper demonstrates empirical results, with substantial GPU memory savings and training speedups across several OGBN benchmarks. The system is implemented with care, theoretical models are included to guide design choices, and the performance analysis is thorough. That said, some aspects of the paper could be strengthened:

First, while training time and GPU memory savings are clearly reported, storage efficiency is not comprehensively analyzed. Metrics like I/O amplification, SSD utilization, or data reuse rate are not presented.

Additionally, the paper compares primarily to DGL’s NeighborSampler and FBL, but omits other caching-aware or storage-optimized samplers such as PyG’s ClusterLoader, Quiver, or recent memory-aware systems.

My third concern is that the connection between the challenges and the proposed caching strategy is underdeveloped. The use of gradient variance to control cache refresh is elegant, but it lacks empirical support or sensitivity analysis. There is no data to show how varying gradient volatility impacts caching performance or hit rates.

Finally, the snapshot metaphor is not well grounded. While the system clearly captures and reuses subgraphs, the concept of a snapshot is never precisely defined, formalized, or tracked across time. A clear abstraction or operational semantics for what constitutes a snapshot would make the system easier to understand and extend.

**Top Reasons To Accept The Paper:**

1- Introduces an efficient storage layout (SEMHS) that reduces I/O overhead for graph sampling.

2- Proposes a flexible, variance-aware cache update strategy (GraphSDSampler) with theoretical underpinnings.

3- Demonstrates empirical gains in memory usage and training speed on large benchmarks.

**Top Reasons To Reject The Paper:**

1- Snapshot concept is metaphorical and under-defined, limiting conceptual clarity.

2- Missing ablations and deeper I/O efficiency analysis, especially for storage-level evaluation.

3- Lack of empirical evidence linking gradient variance to caching performance.

---

### Official Review · Reviewer_msJ4 · 2025-05-17
**Brief summary: This paper proposed GraphSnapShot, an end-to-end system that reorganizes  edge storage into k hop-specific slabs (SEMHS) so that mul#-hop neighbor retrieval becomes  sequen#al I/O, and incorporates GraphSDSampler, a variance-adap#ve controller for mul#-#er  caching across DRAM, HBM, etc., to op#mize hit-rate vs. refresh cost . It also par##ons graphs  into dense and sparse regions—keeping high-degree ver#ces in fast memory and streaming low￾degree nodes on demand—achieving up to 5× loader throughput, 30% end-to-end training #me  reduc#on, and 73% GPU memory savings on standard benchmarks.**

**Confidence:** 3
**Rating:** 6

**Detailed Feedback And Questions For Authors:**

Feedback:
• Comparison with existing works: It would be better to show the advantages of their work
compared to existing works.
• Better illustration: instead of only using formulas, writers can use more figure to
demonstrate how each part works.
• Theoretical Modeling: It would strengthen the work to include an analytical model that
predicts SEMHS I/O savings as a function of graph degree distribution and fan-out.
• Dynamic Graph Scenarios: You can evaluate the cache controller under streaming or
continuously updated graphs to show how quickly it adapts to edge inserts/removals.
• Ablation Clarity: You can provide isolated experiments for SEMHS alone and
GraphSDSampler alone to quantify each component’s individual contribution.

Questions for the Authors
1. Cache Adaptivity: In dynamic graphs, how frequently must you refresh each cache tier
to maintain high hit rates, and what’s the impact on end-to-end latency?
2. Parameter Sensitivity: How sensitive are your results to the choice of fan-out values,
slab sizes, and the gradient-variance metric?

**Top Reasons To Accept The Paper:**

. Interesting Systems Innovation
The SEMHS layout transforms multi‐hop neighborhood sampling into purely sequential
I/O, a substantial departure from random‐access CSR approaches, and directly targets the
dominant bottleneck in large‐scale GNN training .
• Adaptive Caching Framework
GraphSDSampler’s variance‐driven refresh policy unifies diverse cache‐refresh strategies
into a single principled controller, showing clear hit‐rate and latency benefits over fixed
schemes .
• Compelling Empirical Gains
On OGBN and citation‐network benchmarks, the system shows up to 5× loader
throughput improvements, 30% end‐to‐end speedup, and 73% GPU‐memory reduction—
metrics that any graph‐systems paper would be proud to report .
• Practical Impact
The proposed techniques can be integrated into existing frameworks (e.g., DGL or PyG)
with moderate engineering effort, broadening their appeal to practitioners facing I/O‐
bound GNN workloads.

**Top Reasons To Reject The Paper:**

• Lack of comparison to prior works. It would be better to show the advantages of their
work compared to existing works.
• Limited Theoretical Analysis. While the empirical results are strong, the paper lacks a
rigorous analytical model of SEMHS’s I/O cost vs. degree distribution skew, making it
hard to predict performance on unseen graph topologies.
• Engineering Complexity & Overhead
Implementing multi‐slab layouts and a multi-tier cache adds nontrivial complexity. The
paper omits a breakdown of engineering effort or runtime overhead for maintaining
SEMHS structures and adapting cache tiers.

---

### Official Review · Reviewer_Ahq7 · 2025-05-18
**This paper introduces GraphSnapShot, a system that reorganises edge data on SSDs (SEMHS) and drives a multi-level cache with a control-theoretic policy (GraphSDSampler). On OGBN benchmarks the authors report up to 73 % GPU-memory reduction and 30 % training-time speed-up compared with DGL NeighborSampler.**

**Confidence:** 2
**Rating:** 6

**Detailed Feedback And Questions For Authors:**

- Feedback

This workshop paper introduces GraphSnapShot, a promising system for accelerating graph machine learning using the novel SEMHS storage strategy and the adaptive GraphSDSampler caching mechanism, demonstrating notable initial memory savings and speedups. For improvement, the paper should significantly clarify the formulation of its core cache refresh strategy, particularly the different expressions for the refresh ratio $\gamma_l$. Additionally, strengthening the comparative evaluation beyond FBL or clearly detailing the current scope, and providing more specifics on the novelty and implementation of the "graph split" component would be beneficial for the workshop audience.


- Questions

    + For the "graph split," what are the key differentiators of your partitioning strategy compared to existing methods, and could you provide more specifics on the "runtime rule"?

    + While GraphSnapShot shows strong performance on OGBN datasets, what do you foresee as the primary scalability bottlenecks when applying the system to even larger graphs?

**Top Reasons To Accept The Paper:**

-  Novel storage layout (SEMHS) that guarantees at most one sequential read per hop, reducing I/O latency independently of degree distribution.

- Strong empirical results: Demonstrates substantial improvements in memory usage and training time acceleration across OGBN datasets.

- The preliminary experimental results are compelling, demonstrating substantial memory savings  and notable training speedups that indicate high potential for impact in the field of large-scale graph ML, making it a good candidate for the MLArchSys workshop.

**Top Reasons To Reject The Paper:**

- The proposed SEMHS storage strategy offers a novel approach to organize graph data on disk, potentially leading to significant I/O improvements for k-hop neighborhood sampling by enabling sequential reads.

- The "graph split" component, despite its claimed significant impact on reducing random accesses, lacks sufficient detail on its specific implementation and novelty compared to existing graph partitioning techniques, making its contribution difficult to assess.

---

### Official Review · Reviewer_aDvU · 2025-05-19
**Review for GraphSnapShot**

**Confidence:** 2
**Rating:** 6

**Detailed Feedback And Questions For Authors:**

I am unfortunately not an expert in this but:

- How would the method change in dynamic graphs

- How does the adaptive caching affects model convergence

**Top Reasons To Accept The Paper:**

- The paper is well motivated to speed up and reduce the memory usage of GML.

- The authors have performed extensive experiments with 80% memory savings and 4× faster data load compared to baselines

**Top Reasons To Reject The Paper:**

- The methods seems to be applied to static graphs only, which may not be realistic.

- Not sure if this is because I don't work in this area, but the paper presentation seemed to be a bit unreadable for some subsections.